# Current Status of Putative Animal Sources of SARS-CoV-2 Infection in Humans: Wildlife, Domestic Animals and Pets

**DOI:** 10.3390/microorganisms9040868

**Published:** 2021-04-17

**Authors:** Max Maurin, Florence Fenollar, Oleg Mediannikov, Bernard Davoust, Christian Devaux, Didier Raoult

**Affiliations:** 1University Grenoble Alpes, CNRS, Grenoble INP, CHU Grenoble Alpes, TIMC-IMAG, 38000 Grenoble, France; mmaurin@chu-grenoble.fr; 2IHU-Méditerranée Infection, 13005 Marseille, France; florence.fenollar@univ-amu.fr (F.F.); olegusss1@gmail.com (O.M.); bernard.davoust@mageos.com (B.D.); christian.devaux@mediterranee-infection.com (C.D.); 3IRD, AP-HM, SSA, VITROME, Aix Marseille University, 13005 Marseille, France; 4IRD, AP-HM, MEPHI, Aix Marseille University, 13005 Marseille, France; 5Centre National de la Recherche Scientifique, 13005 Marseille, France

**Keywords:** SARS-CoV-2, COVID-19, zoonosis, wild animals, domestic animals, companion animals, pets, animal reservoirs, modes of transmission

## Abstract

SARS-CoV-2 is currently considered to have emerged from a bat coronavirus reservoir. However, the real natural cycle of this virus remains to be elucidated. Moreover, the COVID-19 pandemic has led to novel opportunities for SARS-CoV-2 transmission between humans and susceptible animal species. In silico and in vitro evaluation of the interactions between the SARS-CoV-2 spike protein and eucaryotic angiotensin-converting enzyme 2 (ACE2) receptor have tentatively predicted susceptibility to SARS-CoV-2 infection of several animal species. Although useful, these data do not always correlate with in vivo data obtained in experimental models or during natural infections. Other host biological properties may intervene such as the body temperature, level of receptor expression, co-receptor, restriction factors, and genetic background. The spread of SARS-CoV-2 also depends on the extent and duration of viral shedding in the infected host as well as population density and behaviour (group living and grooming). Overall, current data indicate that the most at-risk interactions between humans and animals for COVID-19 infection are those involving certain mustelids (such as minks and ferrets), rodents (such as hamsters), lagomorphs (especially rabbits), and felines (including cats). Therefore, special attention should be paid to the risk of SARS-CoV-2 infection associated with pets.

## 1. Introduction

Coronaviruses (CoVs) belong to the order Nidovirales, suborder Cornidovirineae, family Coronaviridae, and subfamily Orthocoronavirinae. This subfamily includes four genera termed α-, β-, γ-, and δ-CoVs, corresponding to groups I to IV [1,2]. The term “coronavirus” was coined due to the club-shaped spike projections giving the virus the appearance of a solar corona.

Coronaviruses are found in many vertebrates, although each species has a narrow host spectrum [1]. Bats and birds are considered significant reservoirs of these viruses [3,4,5,6,7]. Coronaviruses mainly infect the respiratory or digestive tracts or both. Systemic infections are rare.

Common human coronaviruses (HCoVs) include two α-CoV (HCoV-E299 and HCoV-NL63) and two β-CoV (HCoV-OC43 and HCoV-HKU1). These viruses are likely to have originated in either bats or rodents [8]. They usually induce mild diseases in humans, such as the common cold. However, severe infections have been occasionally reported in young children, immunocompromised people, and people infected with a specific HCoV-NL63 mutant [9].

Since the 2000s, three β-CoVs of animal origin have led to epidemics in the human population. The first was the severe acute respiratory syndrome coronavirus (SARS-CoV-1), which emerged in humans in 2002–2003 and was considered to originate from horseshoe bats *Rhinolophus affinis* [10]. Approximately 8000 confirmed cases were recorded, with mortality close to 10%. The Middle-Eastern Respiratory Syndrome virus (MERS-CoV) emerged in 2012 [10]. This was also considered to be of bat origin, but humans were probably infected through close contact with dromedaries [11]. Fewer human cases were confirmed, but a 35% fatality rate was reported. The latest outbreak was first detected in December 2019 in Wuhan city, Hubei Province, China. It rapidly turned in to a pandemic (officially recognised as such by the WHO on 11 March 2020 [12]) due to sustained human-to-human transmission of the coronavirus in question. Almost all continents and countries are currently affected by this pandemic. As of 1 February 2021, the WHO reports approximately 102 million confirmed cases of COVID-19, including 2.2 million deaths (https://covid19.who.int/ accessed on 1 February 2021). This coronavirus, first referred to as nCoV-2019, was officially named SARS-CoV-2 by the International Committee for the Taxonomy of Viruses [2]. The WHO proposed the disease name “COVID-19” on 11 February 2020.

COVID-19 is responsible for a mild to severe lower respiratory tract infection in humans [13,14,15]. Following a rapid and robust multiplication of SARS-CoV-2 in the upper and lower respiratory airways, viraemia may spread the virus to many organs. However, the hallmark of COVID-19 is a strong host inflammatory response that may lead to severe acute respiratory syndrome (SARS). Other severe complications can occur, such as thrombotic events due to coagulation disorders.

The current hypothesis for the origin of SARS-CoV-2 corresponds to the zoonotic transmission of this virus to humans, more specifically at the seafood and “wet” live animal wholesale market in Wuhan [16]. Many animal species are susceptible to infection with SARS-CoV-2, SARS-CoV-1, and MERS-CoV [1]. Although horseshoe bats from the species *Rhinolophus affinis* have been proposed as a potential reservoir [17,18,19] and pangolins (*Manis javanica*) as an intermediate host of SARS-CoV-2 [17,19,20,21], the natural zoonotic cycle of this virus remains unknown [22,23].

This review summarises the information that is currently available on the zoonotic nature of SARS-CoV-2 infections, including optimal conditions for the acquisition of this infection, natural and experimental diseases in animals, potential animal reservoirs and intermediate hosts, and modes of transmission of this coronavirus between the human and animal populations. Following the pandemic spread of COVID-19 in humans, new questions have emerged including: Has the diversity of intermediate hosts been increased? What is the risk of reverse zoonosis, that is to say infection of animals from human cases of COVID-19? What is the extent of the spread of SARS-CoV-2 to domestic animals and pets? What is the current role of domestic and companion animals in the SARS-CoV-2 zoonotic risk?

## 2. Potentially Favourable Conditions for the Emergence of SARS-CoV-2

The conditions for the emergence of a new virus in the animal and human populations are varied and complex. They may involve genetic modifications in the virus, leading to an enlarged host range (e.g., changes in interactions between the virus and its eukaryotic cell receptor), changes in the ecosystems (e.g., the density of animal and human populations), or modifications to the interactions between animals (the reservoir and intermediate host) and humans (e.g., lifestyle or eating habits). With regards to SARS-CoV-2, several factors for the emergence of the COVID-19 pandemic have been pointed out.

### 2.1. Viral Genetic Variation

Coronaviruses have the largest viral RNA genomes known to date. It has been suggested that their expansion and selection was enabled by the acquisition of enzyme functions that counter the high error frequency of RNA polymerases [24]. The SARS-CoV-2 genome comprises a large single-stranded positive-sense RNA of 30 kb (29,891 nucleotides) [25]. The G + C content is 38%. The SARS-CoV-2 genome encodes as many as 14 open-reading frames (ORFs), leading to the synthesis of 29 proteins [15,26]. Structural proteins encode the spike (S), envelope (E), matrix (M), and nucleocapsid (N) proteins. Coronaviruses (CoVs) evolve through point mutations and recombination [27]. Spontaneous mutations are favoured by their large RNA genome and low fidelity of their RNA-dependent RNA polymerase (RdRp). Furthermore, the mutation rate of these viruses can be substantially increased under immune pressure (natural immune response of the host or vaccination). RNA recombination events between coronaviruses are facilitated by mixed infections with closely related CoV species in the same host. Recombination between a bat and a pangolin CoV genomes was proposed as a mechanism of SARS-CoV-2 emergence [21,28]. This hypothesis was mainly proposed due to the characterisation of a furin cleavage site unique to SARS-CoV-2 compared to the other Sarbecoviruses [22,29]. Although SARS-CoV-1 lacks the polybasic furin cleavage site found downstream of the RBD in the spike of SARS-CoV-2, such a furin cleavage site (which confers a higher affinity of SARS-CoV-2 for the human ACE2 receptor) was described in many other lineages of coronaviruses and was naturally selected [30]. However, naturally occurring furin cleavage sites have been described in other coronaviruses’ lineages [31,32]. Very recently, Wacharapluesadee and colleagues reported the circulation of a SARS-CoV-2 related coronavirus known as the RacCS203 strain in *Rhinolophus acuminatus* bats from southeast Asia [33]. The RaCS203 genome showed 93.7% identity with the genome sequence of the RmYN02 strain from the *Rhinolophus malayanus* bat. The RaCS203 spike gene was found to be similar to that of RmYN02 and shared part of the furin cleavage site unique to SARS-CoV-2. It is notable that the RBD of RaCS203 indicated that this strain is unlikely to use ACE2 as an entry receptor. Moreover, it was recently reported that the spikes from the Guangdong pangolin coronavirus, closely related to SARS-CoV-2 (a sequence derived from metagenomic but not sequenced from a viral isolate), bind strongly to pangolin and human ACE2 (hACE2) receptors [34]. SARS-CoV-2 and coronaviruses evolve according to the quasi-species model within-host selection of mutants [35,36,37,38]. Taken together, these genetic changes facilitate the efficient interspecies transmission of coronaviruses. Appendix A summarises genome data of SARS-CoV-2 strains isolated from animals, according to GISAID (gisaid.org).

### 2.2. Interactions of Viral Spike with ACE2 and Other Possible Cell Receptors

The S protein of SARS-CoV-2 possesses receptor-binding domain (RBD), antigenic epitopes, and a cleavage site (CS) [25]. The S protein is cleaved by host proteases into S1 and S2 subunits responsible for binding to the host cell receptor and for the fusion of viral and cellular membranes. As for SARS-CoV-2, the eukaryotic cell receptor is the angiotensin-converting enzyme 2 (ACE2). The affinity of the viral S protein (especially the RBD) to the ACE2 receptor highly determines the corresponding host’s susceptibility to infection by this virus. Such ligand-receptor interactions can be evaluated through in silico analyses, in vitro experiments using eukaryotic cells, and in vivo data in animal models or naturally infected animals (Table 1).

In silico analyses of RBD-ACE2 interactions have predicted that humans, some nonhuman primates, bats, pangolins, cats and other felids, dogs, pigs and boars, cattle, sheep, goats, hamsters, ferrets, snakes, whales, and porpoises should be susceptible to SARS-CoV-2 infection (Table 1). Significant RBD-ACE2 interactions were predicted in humans, some monkeys, bats, pangolins, and cats. In cell models expressing the ACE2 receptor of various animal origins, the results largely correlated to those in silico studies (Table 1). In particular, eukaryotic cells expressing hACE2 (Calu3 and Caco2 cells) or monkey ACE2 (VeroE6, FRhK4, and *M. mulatta* ACE2 expressing cells) were susceptible to SARS-CoV-2 infection [48,49]. The same was true for cells expressing ACE2 from cats, dogs, rabbits, pigs, and cows. In vivo, the COVID-19 pandemic has confirmed the susceptibility of humans to SARS-CoV-2 (Table 1). Many monkey species (especially the Rhesus macaque and African green monkey) have developed COVID-like diseases in experimental models [52,74,75,76]. Natural or experimental SARS-CoV-2 infection in other animal species has revealed susceptibility levels that were not fully correlated with in silico and in vitro data (Table 1). For example, cats could be infected naturally or experimentally with SARS-CoV-2 with occasional cat-to-cat transmission of this virus. In contrast, dogs in contact with COVID-19 owners involved with or experimentally infected with SARS-CoV-2 did not develop the overt disease and did not transmit this virus to naïve co-housed dogs.

Beside ACE2, NRP-1 was reported to bind to furin-cleaved substrates, potentiating SARS-CoV-2 infectivity [77,78]. The ACE2 sequences from *Mustelidae* share about 83% amino acid identity with the human ACE2, while the ACE2 sequences from *Neovison vison* and *Mustela lutreola* have 99.51% similarity. In contrast, the NRP-1 protein is much more conserved among species (Devaux et al., manuscript under preparation).

### 2.3. Host Body Temperature

The SARS-CoV-2 spike protein has a broad tropism for ACE2 proteins. However, to better characterise the potential zoonotic repertoire of SARS-CoV-2, it is not enough to look at the compatibility between the virus spike protein and the potential host’s ACE2 receptor. Information about the core temperature of the potential host is crucial. Indeed, the ACE2 receptor in pigs has a greater homology with the hACE2 receptor than with those in cats and ferrets. However, only cats and ferrets are hosts that are susceptible to infection by SARS-CoV-2. The temperature of pigs is estimated at between 39.3 °C and 39.8 °C, while that of cats is 37.8 °C and that of ferrets is between 38.2 °C and 38.8 °C [57,79]. The body temperature of ducks and chickens is estimated at between 40–41.2 °C and 41.6–41.9 °C, respectively, and they do not appear to be sensitive to SARS-CoV-2 [57,79]. Farmed mink (mainly American mink, *Neovison vison*) have also been shown to be vulnerable to SARS-CoV-2 and the body temperature of the European mink (*Mustela lutreola*) has been estimated at between 36.2 and 38.4 °C. Thus, the temperature lability of the SARS-CoV-2 spike protein may limit its host repertoire.

### 2.4. Human and Animal Population Density

A high population density in an animal reservoir or intermediate host will favour the emergence and spread of a new human pathogen. For example, during a large Q fever outbreak in the Netherlands, patients suffering from community-acquired pneumonia caused by *Coxiella burnetii* (the Q fever agent) were more likely to live near farms breeding sheep and goats [80,81]. With regards to wild animals, a study from Dub et al. [82] demonstrated that between 2007 and 2017, an increase in the incidence of tick-borne encephalitis in Finland correlated with the density of white-tailed deer.

Following the emergence of COVID-19, the disease rapidly spread through the human population, due to effective human-to-human transmission. Several studies have demonstrated that significant risk factors for acquiring SARS-CoV-2 infection are related to human-to-human contact rates, including high population density, living in large urban areas, mobility, and low socioeconomic status [83,84,85].

### 2.5. Group-Living and Grooming Habits

It is widely accepted that direct contact is a very effective way of spreading various infectious diseases. Pathogenic microorganisms pass from infected individuals to healthy ones via direct physical contact, sometimes associated with blood or bodily fluids. Such a mode of transmission favours skin and mucosal infections and airborne, vector-borne, and food-borne diseases. Group-living and grooming behaviours are major factors facilitating disease transmission and thus are associated with a significant health risk for the population in question [86,87]. In modeling studies, the spatial aspects were crucial for the evolution of bacterial [88] and viral [89] diseases. The combination of spatial aggregation with frequent grooming behaviours may characterise animal species that can host a transmittable pathogen. Coupled with a genetically highly variable microorganism, this may be a greenhouse for emerging pathogens. Indeed, three large groups of mammals characterised by group-living and intensive grooming behaviour, primates, bats, and rodents [90], are essential sources of zoonotic pathogens in humans.

Most of what is known about social grooming comes from studies of primates [91]. For example, group-living and grooming are of utmost importance for transmitting nematodes in Japanese macaques [92].

Bats have an exceptionally close spatial aggregation, living in colonies. Some species, like vampire bats, demonstrate social grooming and a unique regurgitated food-sharing behaviour that makes them highly exposed to contact-transmissible diseases [93]. Deforestation and anthropised environments are suitable for a wide range of bat species which can find niches that are compatible with their roosting and hunting needs [4,94]. For example, house lights which attract insects at night offer easy prey to insectivorous bats. Houses and barns offer shelter for cave-dwelling bats. Agriculture attracts frugivorous bats.

### 2.6. The Spillover versus Circulation Model

There are currently two models for viral emergence [16,95,96]. The accepted worldwide linear spillover model postulates that an animal reservoir species producing a very high level of the virus must be at the origin of zoonosis [96]. The emergence occurs when the pathogens spill over from the reservoir to inundate other species. This zoonotic pressure triggers a high-frequency infection in humans. Consequently, the animal reservoir species should carry the same virus as the one causing the epidemic. More recently, another model was proposed, based on the idea that there is no need for either a reservoir nor an intermediate species. In this non-linear model, named the circulation model [16,95], there are only susceptible hosts and resistant hosts, regardless of the species (humans are only one species among others). In the circulation model, a virus’s capacity to infect a novel host is determined by the contact between species and minimal receptor compatibility. Many species can be susceptible in the virus circulation model, as demonstrated by SARS-CoV-2.

## 3. Experimental Models for SARS-CoV-2 Infection

Animal models have provided valuable information on viral replication, clinical manifestations, pathological lesions, and inflammatory and immune responses associated with SARS-CoV-2 infection [97]. They have enabled the definition of various animal species’ susceptibility to SARS-CoV-2 and the potential risk of transmission of this virus between animals or between humans and animals. Table 2 summarises this information.

### 3.1. Non-Human Primates

#### 3.1.1. Callithrix Jacchus versus Macaca

Lu et al. [74] compared three models of SARS-CoV-2 infection in nonhuman primates, including Old World monkeys *Macaca mulatta* (Rhesus macaque) and *Macaca fascicularis*, and the New World monkey, *Callithrix jacchus*. Following SARS-CoV-2 illness, these animals displayed fever, weight loss at ten days post-infection (dpi), but no respiratory symptoms. Viral RNA was detected in nasal swabs for the three monkey species, from two dpi (maximum viral load) up to 14 dpi in some animals, with higher viral titres in *Macaca* sp. than in *C. jacchus*. As for the *Macaca* species, viral RNA and infectious virus were detected in the pulmonary tissues. Viral RNA was also detected in many other tissues (including the spleen, gut, and urogenital tract). Severe macroscopic lesions were observed in the lungs. These animals developed a specific antibody response.

In contrast, in *C. jacchus*, the infectious virus was not detected in the pulmonary tissues. No severe macroscopic lung lesions were observed, and the animals did not develop a significant specific antibody response. Overall, *Macaca* sp. were more susceptible to SARS-CoV-2 infection than *C. jacchus*, although none of these animals developed fatal diseases.

#### 3.1.2. Macaca Mullata (Rhesus Macaque)

Other studies evaluated SARS-CoV-2 infection in *M. mullata* [52,74,75,76]. The animals were infected through the intratracheal route [75], the intranasal route [76], the ocular route [104], the intragastric route [104], or a combination of intratracheal, intranasal, ocular and oral routes [52].

The intratracheal inoculation of SARS-CoV-2 in *M. mullata* induced transient fever, reduced appetite, weight loss, dehydration, tachypnea, and a hunched posture [52,75]. Patchy opacities progressing to multiple glass-ground opacities were found in the chest X-rays of some animals [75]. Viral RNA and infectious virus were detected in nasal swabs from 1–2 dpi up to seven dpi and in anal swabs [75]. Pathological findings included consolidation, oedema, haemorrhage, and congestion with interstitial pneumonia [52,75]. Infectious virus was isolated from the trachea, bronchus, and lungs up to 17 dpi [52,75]. Viral RNA was also occasionally detected in the gut and lymphoid tissues and less frequently in other organs (including the spinal cord, heart, skeletal muscles, and bladder) [52]. All animals seroconverted at 10–14 dpi and recovered within three weeks of infection [52].

Similar observations were made in animals infected through the nasal route [76]. Viral RNA and infectious virus were detected in nasal, oropharyngeal, and rectal samples for one to two weeks post-infection. Interstitial pneumonia developed on 5–7 dpi, and viral RNA was detected in the lower respiratory tract and lymph nodes from 5 to 21 dpi. Viral RNA was detected in the lungs and trachea from 3 to 9 dpi and in the lymph nodes from 5–21 dpi. Viral RNA was also detected in other organs. Severe interstitial pneumonia was observed on necropsy.

After intragastric inoculation of SARS-CoV-2, viral RNA was undetectable in tested swabs and tissues collected at seven dpi [104]. In animals euthanised at seven dpi after intraocular infection, viral RNA was primarily detected in the nasolacrimal and ocular system and the upper airways and lungs [104].

In conclusion, the Rhesus macaque model was considered to reproduce the human COVID-19 disease of moderate severity. Severe complications such as SARS and thromboembolic events did not occur, and all animals fully recovered. SARS-CoV-2 infected animals were protected against a second challenge with this virus [105].

#### 3.1.3. African Green Monkey (Chlorocebus Sabaeus)

African green monkeys were also used as a model of SARS-CoV-2 infection [98,99]. These animals were infected through the intranasal and intratracheal routes [98] or through the intranasal route only but using a mucosal atomisation device (MAD) [99]. Most animals experienced transient fever, loss of appetite, lymphocytopenia and thrombocytopenia, and a moderate increase in C-reactive protein. Infected animals then developed respiratory disease and coagulation disorders (including a transient increase in aPTT and circulating fibrinogen levels).

Viral RNA and infectious virus were detected in nasal swabs (from two dpi up to 15 dpi) and rectal swabs (from 2 dpi up to 28 dpi) [99]. Viral RNA was also detected in BAL fluids 3–7 dpi in all animals [98]. In animals euthanised at five dpi [98], viral RNA was detected in the upper and lower respiratory tracts (at high loads) but also in other organs (including the heart, gut, urogenital tract, and central nervous system). Pathological findings mainly included consolidation with hyperaemia and haemorrhage in the lungs. Interestingly, animals euthanised at 34 dpi displayed multifocal chronic interstitial pneumonia, although SARS-CoV-2 was no longer detectable in the lungs [99]. A marked inflammation and coagulopathy in the blood and tissues were also reported [98].

Almost all animals seroconverted [98,99] and developed a specific immune cell response. Three animals which received two SARS-CoV-2 challenges (35 days apart) and which were euthanised 22 days following re-challenge did not display infectious virus or viral RNA in their nasal or BAL fluid samples, indicating immune protection [98].

The African green monkey model was considered to reflect severe human COVID-19 cases more accurately than other non-human primate species.

### 3.2. Bats

Egyptian fruit bats (*Rousettus aegyptiacus*) have been used as a SARS-CoV-2 infection model, although they are genetically distant from horseshoe bats which are considered as a putative reservoir of this virus [56]. Those animals which were infected intranasally remained asymptomatic, but SARS-CoV-2 was detected in the oral cavity up to 12 dpi [56]. Infectious virus was also detected in respiratory tissues and, at a lower level, in the heart, skin, and intestine. Infected animals developed a specific antibody response. The transmission of SARS-CoV-2 from infected to uninfected co-housed bats was demonstrated in this model [56].

### 3.3. Pangolins

Pangolins are a protected animal species. Therefore, no animal model has been developed with these animals. Interestingly, Xiao et al. [21] reported that pangolins carrying a beta-coronavirus were brought into a rescue centre because of signs of respiratory disease, emaciation, lack of appetite, inactivity, and crying. Most of them died within six weeks. Histological findings included diffuse pulmonary alveolar damage of varying severity and lung consolidation in one animal.

### 3.4. Dogs

Beagles intranasally infected with SARS-CoV-2 remained asymptomatic [57]. No viral shedding was detected in nasal and oropharyngeal samples collected 2–6 dpi, while viral RNA was detected in rectal swabs at two dpi in two of five infected animals. Only two animals seroconverted. In one dog euthanised at four dpi, no viral RNA was detected in the collected organs. No infectious virus could be isolated from infected dogs, and no infection occurred in co-housed naïve animals [57]. These data indicate that dogs have a low susceptibility to SAS-CoV-2 infection.

Bosco-Lauth et al. [106] infected three dogs intranasally with SARS-CoV-2. All remained asymptomatic. No viral shedding was detected. On necropsy, no gross lesions were observed. Moderate neutralising antibody titres were detected between 14 and 21 dpi. This study confirmed that SARS-CoV-2 does not replicate in the upper respiratory tract of dogs, and these animals develop low level neutralising antibodies against this virus.

Raccoon dogs were infected with SARS-CoV-2 through the intranasal route [62]. Twenty-four hours later, naïve animals were co-housed with infected ones. Challenged and contact animals remained asymptomatic. Viral RNA and infectious virus were detected in nasal and oropharyngeal samples at 2–4 dpi in most challenged animals. No pneumonic lung lesions were visible and viral RNA was not detected in lung tissue samples on necropsy. A specific antibody response was detected in only about half of infected animals. Two of the three contact animals developed a SARS-CoV-2 infection.

### 3.5. Cats

Shi et al. [57] infected juvenile and subadult (6–9 months old) cats intranasally with SARS-CoV-2. In subadult cats, viral RNA was detected in nasal and soft palate swabs, the trachea, lungs, and small intestines of two animals euthanised at three dpi. Viral RNA was detected in the same samples, with the exception of lung samples at six dpi. Infectious virus was detected in PCR-positive samples except in small intestine samples. Aerosol transmission of SARS-CoV-2 from infected to uninfected cats was demonstrated, although this was inconstant. Infected animals developed a specific antibody response. In juvenile cats (70–100 days old), massive lesions were observed in the nasal and tracheal mucosa and lungs. This study showed that SARS-CoV-2 can replicate in cats and that juvenile cats may develop a more severe infection. In addition, this virus may be transmitted between cats through the aerosol route.

Bosco-Lauth et al. [106] infected five adult cats intranasally with SARS-CoV-2. All inoculated cats remained asymptomatic. Chest X-rays did not reveal any abnormalities. Viral RNA was detected in nasal and oral samples up to 5 dpi. In two cats euthanized at 5 dpi, infectious virus was isolated from the trachea, nasal turbinates, and oesophagus, but not from the lungs or other organs. Pathological findings included moderate rhinitis and tracheitis. In cats euthanised at 42 dpi, mild interstitial lymphocytic pneumonia was observed. Infected cats developed a significant antibody response. Neutralising antibodies were detected as early as seven dpi and reached very high levels by 14 dpi. Two contact cats were co-housed with infected animals challenged two days previously. These contact cats shed infectious virus orally at 24 h post-exposure but for a higher duration than inoculated cats. Upon necropsy at 28 dpi, moderate lymphoplasmacytic rhinitis with rare fibroplasia was observed in the two contact cats. Cat-to-cat transmission of SARS-CoV-2 was also demonstrated by Halfmann et al. [64].

In conclusion, cats appear to be more susceptible to SARS-CoV-2 infection than dogs. The demonstration of viral shedding in infected cats and cat-to-cat transmission raises concern about the potential transmission of SARS-CoV-2 to humans.

### 3.6. Rabbits

Mykytyn et al. [70] infected three-month-old female New Zealand White Rabbits (*Oryctolagus cuniculus*) intranasally with 10^6^ TCID50 SARS-CoV-2. These animals were monitored for 21 days post infection. None of the three inoculated animals showed clinical signs of infection. Although there was high variability between animals, viral RNA was detected in nasal swabs up to 21 dpi, in throat swabs up to 14 dpi, and in rectal swabs up to 9 dpi. Infectious virus was detected in the nose up to 7 dpi, but not in the throat (except in one animal at 1 dpi) and in rectal swabs. All animals monitored for three weeks seroconverted, with serum neutralising antibodies ranging from 1:40 to 1:640.

Three other groups of animals were inoculated intranasally with 10^4^, 10^5^ or 10^6^ TCID50 SARS-CoV-2 and euthanised at 4 dpi. The animals challenged with the highest viral inoculum had a positive viral RNA detection in the nose and throat. Viral RNA shedding was detected in the nose up to 4 dpi and in the throat for 3 dpi in those receiving the medium inoculum. No viral RNA shedding was detected in animals receiving only 10^4^ TCID50 SARS-CoV-2, suggesting a major influence of the infectious viral load on the ability of SARS-CoV-2 to infect and multiply in the upper airway epithelial cells. On necropsy of animals inoculated with 10^6^ TCID50 SARS-CoV-2, viral RNA was detected in nasal turbinates but not in the lung tissue. However, histological lesions included multifocal mild to moderate phagocytic cell infiltration in the lungs, mild peribronchiolar and peribronchial lymphoplasmacytic infiltration, and moderate to severe bronchus-associated lymphoid tissue proliferation.

### 3.7. Mink and Ferrets

Both the ferret (*Mustela putorius furo*) and the mink belong to the Mustelidae family. Ferrets have been used as an animal model of SARS-CoV-2 infection [54,57,100,101]. These animals were infected via the intranasal route [54,57,100,101]. They usually developed mild clinical symptoms 2–8 dpi, including fever, reduced activity, and the occasional cough, but no weight loss [54]. They did not develop SARS and fully recovered within two weeks [54,107]. In nasal washes, viral RNA was detected from 2 dpi up to 20 dpi, and the virus could be isolated from 2 dpi to 8 dpi [54,57,107]. Viral RNA (but not an infectious virus) was also detected from the saliva, urine, feces, and rarely the lungs, kidney, and intestine [54,107]. Viral RNA was not detected in the heart, liver, spleen, pancreas, and brain samples [57]. SARS-CoV-2 was no longer detectable two to three weeks post-infection [57,107].

Pathological findings corresponded to acute bronchiolitis from 4 dpi to 12 dpi [54,107] and mild multifocal bronchopneumonia from 3 to 14 dpi [57,107]. Antibodies against SARS-Cov-2 were detected at 2–3 weeks post-infection, and their titres progressively increased [54,57,107]. Animal-to-animal transmission could be demonstrated either by direct contact or, less efficiently, via the aerosol route [54,55,56].

### 3.8. Mice

Wild mice are considered to be resistant to SARS-CoV-2 infection, supposedly because of the low affinity of their ACE2 receptor to the viral spike protein [72,73,108]. More recently, Zhang et al. [109] reported that BALB/c mice at 12 months old are more sensitive to SARS-CoV-2 than two-month-old mice. Only the former develop pneumonia. However, several adapted mouse models have been developed to better mimic the severity human COVID-19.

Several transgenic mice models expressing the hACE2receptor have been developed [72,110,111,112,113,114,115]. These hACE2 transgenic mice are more susceptible to SARS-CoV-2 infection compared to wild-type animals, with significant viral multiplication and pathological lesions in the lungs. However, only mild clinical symptoms (mainly weight loss) are observed.

A transgenic mice model in which hACE2 is highly expressed through the human cytokeratin 18 promoter was developed [116,117]. In this model, mice succumbed 6 days following intranasal inoculation of 10^5^ PFUs viral load. SARS-CoV-2 was detected in nasal turbinates, lung and brain. A disease of milder severity could be induced by intranasal inoculation of a lower viral load.

Gu et al. [118] selected a mouse-adapted strain of SARS-CoV-2 with higher virulence in these animals. This strain displays several adaptive mutations, including the N501Y mutation located at the RBD of the spike protein.

Dinnon et al. [119] established a recombinant SARS-CoV-2 strain (SARS-CoV-2 MA) able to utilize the mouse ACE2 for viral entry. Using this model, the authors selected a more virulent SARS-Cov-2 strain leading to severe acute lung infection and death in BALB/c mice [120].

Recently, Sefik et al. [121] developed a chronic SARS-CoV-2 infection model in hACE transgenic MISTRG6 mice. These animals developed severe weight loss and lung pathological lesions that persisted for several weeks.

In conclusion, several mice models of SARS-CoV-2 infection have been developed. Some of them leading to severe or even fatal infections are likely more adapted to assess the efficacy of drugs and vaccines against COVID-19.

### 3.9. Hamsters

Syrian [67,68,102] and Chinese [103] hamsters were infected intranasally by SARS-CoV-2. The main clinical symptom was transient but significant weight loss. Other occasional symptoms included lethargy, ruffled fur, a hunched posture, and tachypnea [67,103]. No fatalities were observed [67,103]. Viral RNA and infectious virus were detected in the nasal, oropharyngeal, and tracheal samples at 2 dpi, with rapid clearance within 14 dpi [67,68,103]. The highest viral RNA and infectious virus loads were detected in the lungs [67,68,103]. Lower viral titres were detected in the intestine, salivary glands, heart, liver, spleen, lymph nodes, kidney, brain, and blood, particularly at 4 dpi [67,68]. All hamsters recovered by 14 dpi [67,103]. High serum neutralising antibodies were detected at 7 and 14 dpi [67]. In euthanised hamsters, pathological changes were observed in the nasal turbinate, trachea, and lungs, including lung consolidation and severe pulmonary haemorrhage [67,102,103]. In comparison to Syrian hamsters, pneumonia was milder but more prolonged in Chinese hamsters. Viral transmission to naïve co-housed hamsters was successful, with or without weight loss, but with similar viral shedding and pathological findings in newly infected animals [67,68].

Lee et al. [122] demonstrated that oral inoculation of Syrian hamsters with SARS-CoV-2 resulted in milder symptoms (no weight loss, mild pneumonia) and histological lesions, and lower viral shedding compared to animals infected intranasally.

Osterrieder et al. [123] demonstrated that the severity of SARS-CoV-2 infection in Syrian hamsters depended on the animals’ age. Older hamsters displayed more pronounced weight loss, more severe histological lung lesions, and delayed recovery at 14 dpi than younger animals.

In conclusion, Syrian hamsters are considered a valuable small animal model of SARS-CoV-2 infection, although the animals neither died nor developed severe complications.

### 3.10. Pigs

In two studies, pigs infected through the nasal route with SARS-CoV-2 did not display virus replication (no viral RNA detection) nor an antibody response [56,57]. Vergara-Albert reported that piglets inoculated with SARS-CoV-2 through the intranasal, intratracheal, intramuscular or intravenous routes did not develop infection. However, the animals challenged intramuscularly or intravenously seroconverted 2–3 weeks post-infection [124]. Meekins et al. demonstrated that SARS-CoV-2 could infect swine testicle and porcine kidney (PK-15) cell lines [125]. In contrast, none of the nine pigs infected through the oral, intranasal or intratracheal routes developed clinical signs, viral replication or specific antibody response at 4, 8 and 21 dpi [125]. Viral RNA detection remained negative in blood, lung tissue, and oropharyngeal, nasal, and rectal swabs. Moreover, challenged pigs did not transmit SARS-CoV-2 to uninfected contact animals.

More recently, Pickering et al. [126] infected oronasally 16 domestic pigs. Infected animals were followed for a period of 29 dpi. The only clinical symptoms occurred during the first three dpi, including ocular discharge in all animals, nasal secretion in some, and cough in one. Viral RNA was detected in nasal washes of two pigs at 3 dpi, but viral culture of these samples remained negative. Upon necropsy, no significant pathological findings were observed. However, a submandibular lymph node from one pig tested positive for SARS-VoV-2, both by qRT-PCR and culture. Specific serum antibody titers were found in only two animals at 11–15 dpi, including the SARS-CoV-2 positive animal.

In conclusion, available studies indicate that pigs are poorly susceptible to SARS-CoV-2 infection. They are unlikely to be significant carriers of SARS-CoV-2 nor a significant source of transmission of this coronavirus to humans.

### 3.11. Tree Shrews

Tree shrews infected intranasally with SARS-CoV-2 displayed fever but no other clinical symptoms [127]. Viral RNA was detected up to 12 dpi in the nose, throat, and faeces, and was detected more frequently in younger than adult animals. Viral RNA was also detected in the spleen, intestine, brain, liver, and heart.

### 3.12. Poultry

After an intranasal SARS-CoV-2 challenge, chickens did not display any clinical symptoms, and viral RNA shedding and specific antibody response were not detected [56,57].

The same was true for ducks, turkeys, quail, and geese inoculated intranasally with SARS-CoV-2 [57,128]. These experiments suggest that poultry are not susceptible to SARS-CoV-2 infection and cannot transmit this virus to humans or vice versa.

## 4. Animal Species Susceptible to SARS-CoV-2 Infection, Viral Replication and Viral Spread

### 4.1. Domestic Animals

Questions quickly emerged concerning the potential role that domestic animals infected by SARS-CoV-2 of human or animal origin could play in transmitting the virus to humans or other domestic animal species. This led health authorities to carry out epidemiological investigations, mainly when animals had been in contact with SARS-CoV-2 infected people.

#### 4.1.1. Pets

Overall, approximately 99 pets, including 55 cats, 40 dogs, and one ferret, were reported to be affected by COVID-19 based on positive SARS-CoV-2 RT-PCR (Appendix A). Data on transmission were available for 95 pets. All except one were from the homes of confirmed COVID-19 patients. Most animals were asymptomatic or suffered from mild respiratory symptoms.

Asymptomatic SARS-CoV-2 infection in dogs was first reported on 26 Februray 2020 in Hong Kong [66]. In North America, 21 dogs (including 16 in the United States of America (USA) and five in Mexico) were diagnosed with COVID-19. Five dogs infected in the USA remained asymptomatic. All other dogs exhibited mild respiratory signs. In South America, four dogs were diagnosed with COVID-19 in Argentina. In Asia, thirteen dogs, nine in Hong Kong and four in Japan, were reported to be positive for SARS-CoV-2; all were asymptomatic [66,129]. In Europe, two dogs were diagnosed with COVID-19, one in Denmark connected with a positive mink farm, and one in Italy (Appendix A). Both were asymptomatic.

Thirty-one cats were infected with SARS-CoV-2 in North America, all of which were in the USA. Clinical data were available for thirty cats; ten were asymptomatic, and most others had mild respiratory signs. In South America, six cats, including three in Chile, one in Brazil, and two in Argentina, were diagnosed with COVID-19. In Asia, ten cats (eight in Hong Kong and two in Japan) were reported to be positive for SARS-CoV-2; all were asymptomatic [66,129]. In Europe, eight cats were reported to be positive for SARS-CoV-2 using RT-PCR. Two of the infected cats, in Germany and Russia, were asymptomatic. The six symptomatic cats were infected in Belgium [130], Spain, the United Kingdom, Switzerland, and France (two cases).

Several SARS-CoV-2 antibody seroprevalence studies have tried to evaluate the burden of SARS-CoV-2 infections in pets. Deng et al. [131] tested sera from 485 dogs and 87 cats collected in different parts of China (including Wuhan city) from November 2019 to March 2020 using a specific SARS-CoV-2 ELISA. The dogs included 90 beagles, 147 pets, and 250 street dogs. Cats included 66 pets and 21 street cats. None of these animals displayed anti-SARS-CoV-2 serum antibodies. Another study performed in Wuhan (China) between January and March 2020 showed a seroprevalence of 14.7% in the 102 cats evaluated [132]. A more recent study in Wuhan involving 910 dogs whose sera were collected between January to September 2020 revealed a SARS-CoV-2 seroprevalence of 1.75% [133]. Compared to Deng et al. [131], this new study suggested that the Wuhan dog population could have been exposed to SARS-CoV-2 during rapid human-to-human transmission of this virus. In northern Italy, a study targeting 919 pets at the time the virus was actively circulating in humans showed a seroprevalence of 3.3% (15/451) in dogs and 5.8% (11/191) in cats [65]. Dogs from COVID-19 positive households were significantly more likely to be positive than those from negative households [65]. Lower seroprevalences were reported in Croatia by Stevanovic et al. [134]. From 26 February 2020 to 15 June 2020, 656 dog and 131 cat serum samples collected in three veterinary facilities were tested for the presence of anti-SARS-CoV-2 antibodies. Neutralising antibodies were found in 0.76% cats and 0.31% dogs. More recently, in June 2020, serum samples were collected from 13 dogs and 34 cats in France, two to three months after their owners were diagnosed with COVID-19 [135]. All animals were healthy. Serological testing for SARS-CoV-2 was considered positive when either three microsphere immunoassays (MIA) detecting IgG antibodies against N, S1, or S2 IgG viral proteins were positive, or SARS-CoV-2 neutralising antibodies were detected. Using such stringent criteria, seroprevalence was 21.3% for the 47 animals, 23.5% for cats, and 15.4% for dogs. Using the same criteria, none of the sera collected in 22 dogs and 16 cats from owners with unknown COVID-19 status was found positive.

In conclusion, these data suggest that infections in companion animals might not be unusual, although it appears to be much more clinically significant in cats than dogs. It should be noted that the authorities in Hong Kong, Japan, and the United States have set up a protocol for the reinforced surveillance of domestic carnivores (including dogs, cats and ferrets) in contact with human cases of COVID-19, requiring samples to be taken from these animals. In the United Kingdom, France, Switzerland, Brazil, and Chile, samples are only taken as part of research projects. It is therefore irrelevant to compare the numbers of cases across countries. In addition, COVID-19 should be added to the list of diseases potentially transmitted from uncommon pets. According to natural and experimental SARS-CoV-2 infection, special attention should be paid to ferrets and other mustelids, some rodents such as hamsters, and lagomorphs such as dwarf rabbits. Viral RNA shedding was detected in nasal and oral samples up to 2–3 weeks following SARS-CoV-2 infection in some of these animals, and transmission between co-housed animals was demonstrated [54,57,64,68,102,103] (see Figure 1 and Table 2).

#### 4.1.2. Other Domestic Animals

To date, the SARS-CoV-2 virus has not been detected in other domestic animals in natural conditions. Experimental studies by several research teams on poultry, ducks, turkeys and pigs have shown no sensitivity of these species to SARS-CoV-2 [56,57]. Therefore, these farm animals are considered unlikely to transmit COVID-19 to humans or vice versa. In contrast, it has been shown in experimental models that rabbits are susceptible to SARS-CoV-2 infection [70].

### 4.2. Captive Wild Animals in Farms

Mink are non-domestic farm animals raised primarily for their fur. Because of its superior pelage, the American mink (*Neovison vison*) is the preferred species. COVID-19 was first detected in a mink farm in the Netherlands on 23 April 2020. COVID-19 was then detected in mink farms in Denmark in mid-June, in Spain at the beginning of July, in the United States and Italy in August, in Sweden in October, then in Greece, in France, in Poland, and Lithuania in November, and Canada in December (https://www.oie.int/en/ accessed on 5 February 2021) [60]. As of 5 January 2021, farmed mink positive for SARS-CoV-2 had been detected by RT-PCR in several countries, including the Netherlands (69 mink farms), Denmark (290), Spain (1), the United States (17), Sweden (13), Italy (1), Greece (22), France (1), Poland (8), Lithuania (2), and Canada (2). SARS-CoV-2 infections in mink may be asymptomatic or manifest as loss of appetite, digestive or respiratory signs, up to death [59]. Necropsies of dead mink revealed acute interstitial pneumonia in almost all of the mink examined [59].

SARS-CoV-2 was first introduced in mink farms by humans and then evolved, circulating widely among the mink for several weeks before detection [61]. Despite stringent measures, transmission occurred between mink farms with unknown transmission modes [61]. Analysis and comparison of whole genomes of SARS-CoV-2 show that humans were infected with strains with an animal sequence signature, providing evidence of animal-to-human transmission of SARS-CoV-2 on mink farms [61].

Furthermore, on 11 May 2020, a variant of SARS-CoV-2 with mutations in the spike protein was identified in Denmark from mink in five mink farms in North Jutland and in twelve people. This led the Danish authorities to decide to slaughter all mink [136]. The virus may have continued to circulate in mink farms for a long time, representing a risk to public health. The chance that mink could become a reservoir of SARS-CoV-2 should not be neglected in areas with high density of mink farms.

### 4.3. Captive wild Animals in Zoos

Several animals in zoos have contracted COVID-19 (Table 3). They are almost all part of the Felidae family. Overall, seven lions, *Panthera leo*, have been reported to be infected with SARS-CoV-2 (three at the Bronx Zoo in New York and four at the Barcelona Zoo in Spain), as well as seven tigers, including *Panthera tigris jacksoni* and *Panthera tigris altaica* (four at the Bronx Zoo (New York city, NY, USA) and three at the Knoxville zoo (Knoxville, TN, USA), three snow leopards, *Panthera uncia* (Jefferson Zoo in Kentucky, USA), and one cougar, *Puma concolor* (Johannesburg zoo in South Africa). Another Hominidae, the western lowland gorilla, *Gorilla gorilla*, has also been infected with SARS-CoV-2. Indeed, three western lowland gorillas out of eight co-housed together in a troop at the San Diego Zoo in California were confirmed as being positive for SARS-CoV-2. Almost all the animals were symptomatic and presented with mild respiratory signs such as coughing and wheezing (Table 3). All recovered. It was likely that animals were contaminated from a staff member of the zoo infected with SARS-CoV-2. However, it is possible that after contamination of one of the Felidae by a staff member of the zoo, the Felidae contaminated the other animals.

### 4.4. Non-Captive Wild Animals

Bats (*Chiroptera* order of mammals) include more than 1300 species spread worldwide, with the exception of Antarctica. However, their geographic distribution is species dependent. Bats contribute to the evolution and dissemination of alpha-coronaviruses and beta-coronaviruses [137]. They are the preferred hosts for multiple virus strains and are probably preferential hosts for alpha-coronaviruses and beta-coronaviruses [7]. It is thought that many human coronaviruses may be of bat origin [138], although HCoV-OC43 probably, passed to humans from rodents [139]. Researchers speculate that all four human coronaviruses that cause the common cold emerged as human pathogens over several centuries and likely caused pandemics at the time of the transition [140]. Molecular clock analysis of the spike gene sequences of HCoV-OC43 suggests a relatively recent zoonotic transmission event. It dates the separation from its ancestor to around 1890 [141], which coincides with the 1889–1890 flu pandemic, also known as the “Asian flu” or “Russian flu”.

Although direct transmission of the coronavirus from bats to humans is possible, molecular data suggest the presence of another (intermediate?) host that also contributed genetically in a SARS-CoV2 structure [20]. Differences in the whole genome sequence of SARS-CoV-2 and pangolin-CoV indicate that the latter cannot be considered an immediate ancestor of the former [18]. Moreover, an ecological link between bats and pangolins is not easily to reconstruct. A possible connection between bats and humans may be constituted via bat-hunting animals.

Bats have few natural predators. Owls, hawks, and snakes are reported to eat bats. Birds are the usual hosts of gamma- and delta-coronaviruses. No evidence of beta-coronavirus in wild birds has been reported, with the exception of one study in Brazil detecting beta-coronavirus RNA in wild birds preying on bats [142]. Moreover, the predicted affinity of bird ACE2 receptors to bind to SARS-CoV-2 is very low. An almost identical situation holds for reptiles [143].

Cebidae New World monkeys have been repeatedly reported to prey on bats [144,145,146]. Similar behaviour has been noted in *Cercopithecus* in Kenya and Tanzania [147].

Other bat-hunting animals include raccoons [148,149], otters [150], mink [151], sable [152], long-tailed weasels [153], and Siberian weasels [154]. The Siberian weasel, also referred to as a kolonok, is widely distributed across north-eastern Asia, including a vast region in eastern China, extending from Heilongjiang in the north to Yunnan in the south. It largely inhabits forest and forest-steppe areas, often settling near rivers. The basis of its diet in natural landscapes is small mammals and birds. However, in winter, when the prey is scarce, kolonok may often hunt on bats [154]. Because they are Mustelids, which are a priori susceptible to coronavirus infection, kolonoks may be an interesting candidate for the link between coronavirus-hosting bats and sensitive humans or mink farms. Kolonoks are also hunted for their perfect fur, and wild carnivores may come into contact with minks in farms when trying to steal food [155].

### 4.5. World Organisation for Animal Health (OIE) and SARS-CoV-2 in Animals

The OIE recently issued a technical factsheet on infection with SARS-CoV-2 in animals [156]. This factsheet emphasizes the high susceptibility to SARS-CoV-2 of cats and other felines (tigers, lions, leopards, and pumas), white tailed deer, Golden Syrian hamsters, Egyptian fruit bats, gorillas, marmosets, and macaques. The OIE advocates that SARS-CoV-2 infected people (or people suspected to be infected with this virus) should restrict contact with mammalian animals, including pets. Likewise, animals with suspected or confirmed SARS-CoV-2 infection should remain separated from other animals and humans. Further information from OIE can be obtained from the WAHIS portal for Animal Health Data (https://www.oie.int/en/animal-health-in-the-world/wahis-portal-animal-health-data/ accessed on 5 February 2021).

## 5. Potential Interspecies Transmission of SARS-CoV-2

### 5.1. Transmission between Animals

The risk of transmission of SARS-CoV-2 from one mink farm to another via mink or personnel movements is high. During the first SARS-CoV-2 outbreak in mink in the Netherlands, samples from 11 cats were analysed. They were all RT-PCR negative, but three had positive SARS-CoV-2 serology. One case of an infected dog was also linked to a SARS-CoV-2 outbreak in a mink farm. Thus, there is a risk of transmission from minks to dogs and cats.

On 13 December 2020, the National Veterinary Services Laboratories (NVSL) of the United States Department of Agriculture (USDA) confirmed (using real-time RT-PCR and sequencing of a nasal swab) a SARS-CoV-2 infection in wild mink caught near a COVID-19 infected mink farm in Utah (USA) [157]. More recently, natural SARS-CoV-2 infections were further reported in two wild American minks in the Valencia Community, Spain [158]. SARS-CoV-2 Infection was confirmed by viral RNA detection in mesenteric lymph nodes. These animals were trapped far away from the nearest fur farm, suggesting other sources of infection, including SARS-CoV-2 contaminated wastewater.

Natural and experimental infections have shown that several animal species are susceptible to SARS-CoV-2, including non-human primates, cats, dogs, raccoon dogs, bats, pangolins, felids, mustelids, rodents, and lagomorphs. The number of animal species susceptible to this virus is probably much more extensive. There is, therefore, a real concern about SARS-CoV-2 transmission within and between these species. Transmission of this coronavirus between the domestic and wild animal populations should be specifically evaluated.

### 5.2. Transmission between Humans and Domestic, Farm, or Zoo Animals

The transmission of SARS-CoV-2 from humans to a domestic animal species appears to be rare and sporadic, considering the high-level circulation of the virus in the human population (Appendix A). This transmission is mainly linked to significant contact between animals and humans in closed or confined environments. The transmission of SARS-CoV-2 to pets from humans has been reported primarily in relation to cats and dogs. One case of transmission from a human to a domestic ferret has also been reported. Several zoo animals, mainly Felids (lions, tigers, cougars, and snow leopards), have been infected by staff members of zoos (Table 3). More recently, SARS-CoV-2 transmission from a human to a gorilla has also been reported.

To date, there is no scientific evidence of transmission of SARS-CoV-2 from pets to humans, including cats and dogs. Thus, owners should not abandon their pets or compromise their welfare [159]. However, they should monitor their pets to detect any health problems and apply the required hygiene and biosafety rules. They should particularly avoid contact between the ill pet and other animals and humans. With regards to captive wild animals, the risk of transmission of SARS-CoV-2 from mink to humans is high. Hygiene and biosafety measures should be reinforced.

### 5.3. Transmission between Humans and Wild Animals

According to the spillover model, the animal reservoir and intermediate hosts of SARS-CoV-2 remain to be fully characterised. Direct transmission of this coronavirus from bats, pangolins, or other animals to humans has not been demonstrated. This was recently confirmed by the first WHO committee site visit to Wuhan. As indicated above, many wildlife species are susceptible to SARS-CoV-2, a result which is more compatible with the circulation model. The transmission of COVID-19 from humans to wild animals and vice-versa should be monitored. The USDA report of a first case of SARS-CoV-2 infection in a wild mink in Utah (USA) [157] indicates that wildlife reservoirs of SARS-CoV-2 might emerge in many susceptible species.

Deng et al. [131] attempted to identify potential intermediate wildlife hosts of SARS-CoV-2 that could have transmitted this virus to humans. Sera from 313 animals corresponding to 21 species were collected in various geographic locations in China from November 2019 to March 2020. The tested species included mink (*n* = 91), foxes (89), camels (*n* = 31), pangolins (17), giant pandas (14), masked civets (10), alpacas (10), bears (9), bamboo rats (8), tigers (8), a few peacocks, rhinoceros, yellow-throated marten, leopard cats, red pandas, ferrets, porcupines, and one eagle, jackal, weasel, and wild boar. Using a specific SARS-CoV-2 ELISA previously validated using positive and negative sera, none of the tested animals displayed anti-SARS-CoV-2 serum antibodies.

## 6. Spatial Aggregation of Susceptible Hosts Increases the Risk of SARS-CoV-2 Variant Selection

Data from the previous sections of this review lead us to propose a simple model for the spread of SARS-CoV-2 variants in humans. Firstly, the RNApol RNA-dependent induced errors, the existence of viral quasispecies, host selection pressure, viral fitness, and the number of passages from one individual to another encourage different mutations in a viral strain. It was shown that the initial rapid growth process of a virus within a cell leads to a sharp increase in diversity [160]. Thus, the higher the circulation of the virus, i.e., the frequency of transmission from one individual to another, the greater the genetic variability of the circulating virus. Of course, this depends on the capacities of the virus itself to accumulate mutations. Facilitated by spatial aggregation and frequent grooming behavior in some animals, the accumulation of new mutations may at some point lead to a fortuitously adapted viral genotype capable of infecting previously unsusceptible hosts through a spillover effect. Host jumps and associated genetic diversity can also arise through various ecological and evolutionary mechanisms [161].

Hence, such group-living mammals with high spatial aggregation and frequent grooming behaviour, such as bats, primates, and rodents may represent a potential incubator for novel zoonotic infections. We hypothesise the epidemiological model of the emergence of the SARS-CoV-2 virus from bat coronaviruses (Figure 2). It seems that SARS-CoV-2 is closely related to the MN996532_raTG13 and RmYN02 coronaviruses from the Chinese horseshoe bats *Rhinolophus affinis* and *Rhinolophus malayanus*, respectively [19,162].

The role of the intermediate animal host, whose coronaviruses might have taken part in recombination resulting in the emergence of SARS-CoV-2, is not yet widely accepted [20,163]. Owing to the paucity of our knowledge on wildlife-associated coronaviruses, recombination events may happen at any stage and are not discussed in the present model (Figure 2).

## 7. What Remains Unexplored

The above information allows us to partly answer the questions raised in the introduction. Reverse zoonosis has been demonstrated for some pets (cats, dogs, and one ferret), and in captive wildlife in farms (minks) or zoos (lions, tigers, leopards, one cougar, and gorillas).

Experimental models have shown that most domesticated animals are not susceptible to SARS-CoV-2 infection, including cattle, sheep, goats, and pigs. Only rabbits were susceptible to this coronavirus. As for pets, dogs were resistant to ARS-Cov-2 infection while cats developed mild symptoms and presented transient viral shedding from the upper respiratory tract. Rabbits, ferrets, and hamsters developed severe lung involvement and systemic infection with viral shedding from the upper airways for between one and three weeks. In some animals, SARS-CoV-2 was also detected in their faeces and urine. These animals thus appear to be the most vulnerable species to reverse zoonosis, although their role in the transmission of SARS-CoV-2 to humans has not yet been demonstrated.

Some aspects of the zoonotic nature of COVID-19 remain to be explored and clarified. In the spillover model, defining the natural SARS-CoV-2 animal reservoirs and susceptible range of species is of utmost importance to understanding the mechanisms of emergence and spread of this virus. There are no specific animal reservoirs or intermediate hosts in the circulation model, only susceptible and resistant hosts [16,95]. Controlling COVID-19 in humans and animals is highly dependent on the model that applies. Identifying the correct model would help define the risk of modifications in the SARS-CoV-2 animal reservoirs and the diversity of potential intermediate hosts. A primary goal would be to avoid or limit the extent of future zoonotic epidemics with SARS-CoV-2 or other coronaviruses.

Interactions between humans and animals are permanent, although occur more frequently with domestic animals. These pose the risk of transmission of the virus from humans to animals and vice versa. This cycle could prolong the COVID pandemic and lead to the constitution of new animal reservoirs. Ultimately, SARS-CoV-2 could spread in particular ecological niches and reappear regularly. The human population is currently the most affected by the COVID epidemic. Therefore, it is necessary to develop tools and strategies to assess the spread of SARS-CoV-2 in domestic and wild animal populations.

Particular emphasis should be given to pet animals. Some of them (such as cats) can be infected by their owners and can potentially transmit the disease to other animal and human hosts. Pets also often come into contact with wildlife, which can be another source of SARS-CoV-2 infection. Although the current data is fairly reassuring with a low number of infections in domestic animals and pets, the actual risk of SARS-CoV-2 transmission from pets susceptible to this virus to humans and vice-versa needs more precise evaluation.

It remains unclear whether COVID-19 is a long term or short term immunising disease. This information is essential in humans as well as in animals. The duration of virus carriage in infected hosts will condition the risk of the disease being transmitted to humans or animals. A global immunisation strategy has been developed for humans. Vaccination should be considered and evaluated in animals, at least for some farm animals and pets. The risk associated with owning a pet should also be assessed, especially for animals which are highly susceptible to SARS-CoV-2, such as ferrets, hamsters, and rabbits.

Several mutations in SARS-CoV-2, especially those in the RBD of the spike protein, have raised concerns about the higher transmissibility or virulence of this coronavirus. The same holds for animals. Such mutations could potentially also change the range of susceptible animal species. As mentioned above, the susceptibility of animal species is highly dependent on the RBD-ACE2 interactions.

## 8. Conclusions

COVID-19 is the first pandemic of the 21st century. It has a significant impact in terms of human and animal health and the economic burden. It has profoundly changed our lifestyles and our conception of the risk associated with infectious diseases. Although human-to-human transmission of SARS-CoV-2 is currently the most predominant mode of transmission of this virus, the zoonotic origin of COVID-19 is undoubted. Available genetic and epidemiological data strongly indicate that bats are likely to have been involved in the emergence of SARS-CoV-2 from an ancestor coronavirus. However, the natural reservoirs and cycle of this virus remain to be elucidated. In silico, in vitro, and in vivo studies have led to an understanding of some of the factors involved in the susceptibility of a specific host to SARS-CoV-2 infection. However, these data do not lend themselves to assessing the role of a particular animal species in the emergence and spread of this coronavirus. The extent of the COVID-19 pandemic in wild animals is challenging to evaluate and remains largely uncharacterised. Although most domestic animals do not appear to be highly susceptible to SARS-CoV-2, the risk associated with pet ownership should be better defined. Many animals (including some mustelids, rodents, and lagomorphs) are highly susceptible to SARS-CoV-2. Finally, since a large proportion of the human population has been or will be infected with SARS-CoV-2, there is a significant concern about reverse zoonosis, i.e., the spread of this virus from infected humans to naïve domestic or wild animals. The current situation of COVID-19 is rapidly evolving, which justifies monitoring this pandemic both in the human and animal populations. Prophylactic measures (avoiding close contact and vaccination) currently considered for humans should also be considered for some animals. COVID-19 is paradigmatic of the need for a one-health approach to control zoonotic diseases.

## Figures and Tables

**Figure 1 microorganisms-09-00868-f001:**
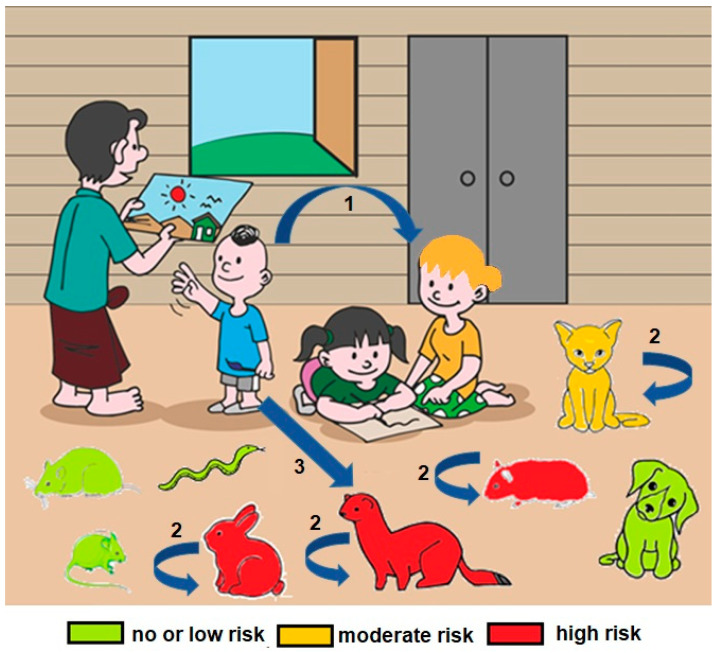
SARS-CoV-2 zoonotic risk associated with exposure to pets. The susceptibility of pets to SARS-CoV-2 infection, and therefore the potential risk of transmission of this virus from these animals to humans, can be evaluated as nul or low (green animals), medium (yellow) or high (red). The arrows and numbers indicate the currently demonstrated transmission chain of SARS-CoV-2: (1) from human-to-human; (2) from animal-to-animal within a specific animal species (cats, hamsters, and ferrets); and (3) from human-to-animal (cats and ferrets).

**Figure 2 microorganisms-09-00868-f002:**
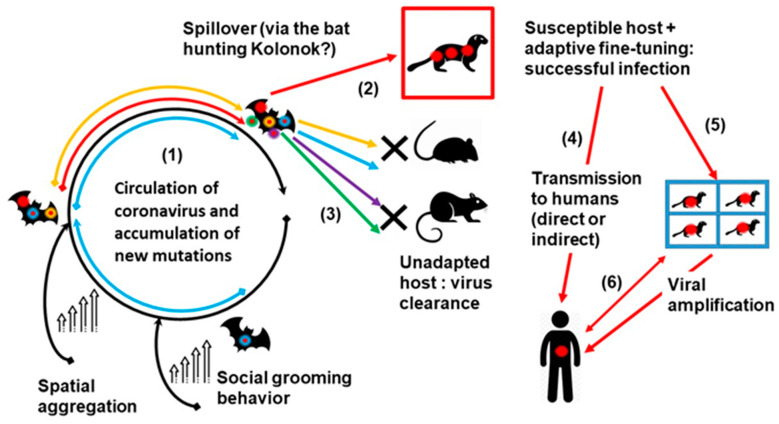
Epidemiological schema of SARS-CoV-2 virus emergence from bat coronaviruses. This figure represents an hypothesis of SARS-CoV-2 emergence and spread, including the following steps: (1) the circulation of coronaviruses in bats, which are animals with spatial aggregation and grooming behavior, can lead to the emergence of new viral genotypes (including SARS-CoV-2, red star) via mutations and recombinations; (2) a given animal species (e.g., a bat predator such as the kolonok) might be infected by SARS-CoV-2, whereas (3) other animals (e.g., mice and rats) remain unsusceptible to infection by any of the new genotypes (unadapted hosts); (4) the SARS-CoV-2 infected animal species may transmit this virus to humans through direct contact or indirectly (e.g., via the consumption of contaminated food products), or (5) after amplification of the virus in other animal hosts; (6) infected humans may transmit the new coronavirus to susceptible farm animals (e.g., the minks) and pets, themselves becoming potential sources of human infections.

**Table 1 microorganisms-09-00868-t001:** Angiotensin-converting enzyme 2 (ACE2) ability to be recognized by SARS-CoV-2 (Wuhan Hu1 strain/G clade).

Species (Human ACE2 and ACE2 Orthologs)	In Silico, Data from [39,40,41,42,43,44,45,46,47,48] ^1^	In Vitro (in Cells), Data from [48,49], S = Susceptible, NS = Non Susceptible, to SARS-CoV-2 Infection	In Vivo, S = Susceptible, NS = Non Susceptible, to SARS-CoV-2 Infection
Human (*Homo sapiens*)	Yes (+++)	S for Calu3 cell sand Caco2	COVID-19 outbreak [15,50,51]
Monkeys (*Gorilla gorilla gorilla, Macaca mulatta; Pan troglodytes*, *Pongo abelii, Papio Anubis*)	Yes (+++)	S for VeroE6 cells and FRhK4 cells, and for HEK293 cells expressing the monkey (*M. mulatta*) ACE2	S (COVID-19-like signs) [52,53]
Monkeys (*Callithrix jacchus/marmoset, tufted capuchin, squirrel monkey))*	Undetermined (− to ++)	S for HeLa cells expressing the monkey (*marmoset*) ACE2	
Ferret (*Mustela putorius furo*)	Yes (++)		S (COVID-19-like signs) [54,55,56,57,58]
Mink (*Mustela lutreola; Neovison vison*)			S (COVID-19-like signs) Mink-to-mink transmission and mink-to-human transmission reported [59,60,61]
Ermine/short tailed weasel (*Mustela erminea*)	Yes (++)		
Raccoon dog (*Nyctereutes procyonoides*)			S (with minor clinical signs) Raccoon dog to raccoon dog transmission [62]
Civet (*Paguma larvata*)	Undetermined (− to ++)		
Pangolin (*Manis javanica*)	Yes (+++)		
Pangolins (*Manis pentadactyla, Smutsia temminckii; Phataginus tricuspis*)	No (−)		
Bats (*Rhinolophus sinicus; Rhinolophus pearsonii; Rhinolophus macrotis*)	Yes (+++)		S [56]
Bats (*Rhinolophus ferrumequinum*, *Myotis*)	No (−)		
Bat (Desmodus rotundus)	No (−)		
Camel (*Camelus dromedarius*)	Undetermined (− to ++)		
Lion (*Panthera leo*)			S [63]
Tiger (*Panthera tigris*)	Yes (++)		S [45,63]
Cat (*Felis catus*)	Yes (+++)	S for CRFK cells and HEK293 cells expressing the cat (*F. catus*) ACE2	S (COVID-19-like signs) Cat-to-cat transmission and Human -to- cat transmission have been reported [57,64,65]
Dog (*Canis lupus familiaris, Canis lupus dingo*)	Yes (++)	S for HEK293 cells expressing the dog (*C. lupus*) ACE2	S, yet the virus replicates very poorly (Human -to- dog transmission has been reported) [57,65,66]
Hamster (*Mesocricetus auratus*)	Yes (++)		S (COVID-19-like signs) [67,68,69]
Rabbit (*Oryctolagus cuniculus*)	Yes (++)	S for HEK293 cells expressing the rabbit (*O. cuniculus*) ACE2	S. Infected animals produce virus [70]
Pig (*Sus scrofa domesticus*)	Yes (++)	S for PK-15 cells and HeLa cells expressing the pig (*S. scrofa*) ACE2	S, yet the virus replicates very poorly [56,57]
Boar (*Sus scrofa*)	Yes (++)		
Cow (*Bos taurus*)	Yes (++)	S for HeLa cells expressing the cow (*B. taurus*) ACE2	S, yet the virus replicates very poorly Cow-to-cow transmission [71]
Buffalo (*Bubalus bubalus*)	Yes (++)		
Goat (*Capra hircus*)	Yes (++)		
Sheep (*Ovis aries*)	Yes (++)		
Rats (*Rattus rattus, Rattus norvegicus*)	Undetermined (− to +)	NS for HEK293 cells expressing the rat (*R. norvegicus*) ACE2	
Mouse (*Mus musculus*)	No (-)	NS for HeLa cells expressing the mouse (*M. musculus*) ACE2	NS, (hACE2 humanized mice are susceptible to infection and show (COVID-19-like signs) [72,73]
Pigeon (*Columbia livia*)	Undetermined (− to +)		
Hen (*Gallus gallus*)	Undetermined (− to +)		
Chiken			S, yet the virus replicates very poorly [57]
Duck			S, yet the virus replicates very poorly [57]
Turtle (*Pelodiscus sinensis; Chrysemys picta bellii*, *Chelonia mydas*)	Undetermined (− to ++)		
Snake (*Ophiophagus hannah*)	Undetermined (− to +)		
Snake/Pallas pit viper (*Protobothrops mucrosquamatus*)	Yes (++)		
Frog (*Xenopus tropicalis*)	No (−)		
Whale/Yangtze finless porpoise (*Neophocaena asiaeorientalis asiaeorientalis)*	Yes (++)		

^1^ These various studies defined an arbitrary cut-off based on the number of conserved amino acids (variable from one study to another) considered critical for interaction with the SARS-CoV-2 spike. The results are generally consistent; when predictions differ, it is summarized as undetermined.

**Table 2 microorganisms-09-00868-t002:** Experimental models of SARS-CoV-2 infection. The route of infection was intranasal, unless otherwise specified.

Animal	Clinical Symptoms	Viral RNA Detection	Infectious Virus Detection	Pathological Lung Lesions	Other Organs Involved	Specific Antibody Response	Transmission to Contact Animals	References
*Callithrix jacchus* *Macaca fasicularis* *Macaca mulatta*	Fever, body weight loss	Nose, lower viral load in *C. jacchus*	Lung, for Macaca only	Interstitial pneumonia, more severe in M. mulatta	Spleen and lymph nodes for Macaca only	Only for Macaca	ND	[74]
Rhesus macaque (*M. mullata*)	Fever, loss of appetite and reduced activity	Nose and oropharynx, than rectal swabs, lungs, lymph nodes	Rectal swabs	Severe interstitial pneumonia	Brain, spinal cords, kidney, liver, spleen, heart, intestine and testicle			[76]
Rhesus macaque (*M. mullata*) IT	Fever, bodyweight loss, dehydration, tachypnea	Nose, oropharynx, anal swab, lungs, gut, lymphoid tissues, and rarely other tissues	Nose, oropharynx, anal swab, trachea, bronchus, lungs	Severe interstitial pneumonia	Gut, lymphoid tissues, spinal cord, heart, skeletal muscles and bladder	Yes	ND	[75]
African green monkey (IT and IN; or IN with MAD)	Fever, loss of appetite, pneumonia, and coagulation disorders	Nose, rectal swab, BAL fluid, lungs	Nose, rectal swab	Multifocal chronic interstitial pneumonia	Lymphoid tissue, heart, gut, bladder, brain, and eyes	Yes	ND	[98,99]
Egyptian fruit bats (*Rousettus aegyptiacus*)	None	Oral cavity, trachea, lungs, lymph nodes, heart, skin, duodenum, adrenal gland tissues	Nose, trachea			Yes	Yes	[56]
Dogs	None	Rectal swabs at 2 dpi only	No	No	No	Yes	No	[57]
Raccoon dogs	None	Nose, oropharynx	Nose, oropharynx	No	No	Yes	Yes	[62]
Cats	Mild or no symptoms	Nose, soft palates, tonsils, trachea, lungs, small intestine	Nose	Severe lung lesions		Yes	Yes	[57,64]
Rabbits	No symptoms	Nose, throat, feces	Nose	Mild to moderate phagocytic cells infiltration	No	Yes	ND	[70]
Ferrets	Fever, reduced activity, occasional cough	Nose, saliva, urine, feces, and rarely the lungs, kidney, and intestine	Nose only	Acute bronchiolitis, mild multifocal bronchopneumonia, and severe lung lesions		Yes	Yes	[54,57,100,101]
Syrian and Chinese hamsters	Body weight loss	Nose, oropharynx, trachea, and many other tissues	Nose, oropharynx, trachea	Severe lung lesions (milder but more prolonged in Chinese hamsters)		Yes	Yes	[67,68,102,103]

IT: intratracheal; IN: intranasal; MAD: mucosal atomization device; BAL fluid: bronchoalveolar lavage fluid; ND: not done; dpi: days post-infection.

**Table 3 microorganisms-09-00868-t003:** Reports of zoo animals diagnosed with COVID-19 using SARS-CoV-2 RT-PCR.

Start Date of the Outbreak	Zoo Location	Animals	Clinical Symptoms	Sources
03/27/20	WCS Bronx zoo, New York, USA	4 tigers ^1^ (*Panthera tigris*) out of 5	Respiratory signs	https://www.oie.int/wahis_2/public/wahid.php/Reviewreport/Review?page_refer=MapFullEventReport&reportid=33885 (accessed on 5 February 2021) https://promedmail.org/promed-post/?id=7191352 (accessed on 5 February 2021)
03/27/20	WCS Bronx zoo, New York, USA	3 lions ^1^ (*Panthera leo*) out of 3	Respiratory signs	https://www.oie.int/wahis_2/public/wahid.php/Reviewreport/Review?page_refer=MapFullEventReport&reportid=33885 (accessed on 5 February 2021) https://promedmail.org/promed-post/?id=7191352 (accessed on 5 February 2021)
10/12/20	Knoxville, Tennessee, USA	3 tigers (*Panthera tigris*) out of 3	Respiratory signs	https://www.oie.int/wahis_2/public/wahid.php/Reviewreport/Review?page_refer=MapFullEventReport&reportid=36433https://promedmail.org/promed-post/?id=7915683 (accessed on 5 February 2021)
11/27/20	Jefferson Kentucky, USA	3 snow leopards (*Panthera uncia*) out of	Respiratory signs	https://www.oie.int/wahis_2/public/wahid.php/Reviewreport/Review?page_refer=MapFullEventReport&reportid=37147 (accessed on 5 February 2021)
07/17/20	Johannesburg, South Africa	1 cougar (*Puma concolor*) out of 2	NA	https://www.oie.int/wahis_2/public/wahid.php/Reviewreport/Review?page_refer=MapFullEventReport&reportid=35399 (accessed on 5 February 2021)
12/10/20	Barcelona, Spain	4 lions (*Panthera leo*)	Respiratory signs	https://promedmail.org/promed-post/?id=8002466 (accessed on 5 February 2021)
01/06/21	San Diego, California, USA	3 gorilla (*Gorilla gorilla gorilla*) out of 8	Respiratory signs for 2 of them	https://www.oie.int/wahis_2/public/wahid.php/Reviewreport/Review?page_refer=MapFullEventReport&reportid=37553 (accessed on 5 February 2021)

^1^ housed in 2 separate enclosures; it is assumed that an asymptomatic zoo employee infected the animals.

## Data Availability

Not applicable.

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
