# Peer review of "Current Status of Putative Animal Sources of SARS-CoV-2 Infection in Humans: Wildlife, Domestic Animals and Pets"

_microorganisms, 2021, doi:10.3390/microorganisms9040868_

Round 1

Reviewer 1 Report

In this review article, Max Maurin and colleagues summarized the current status of putative animal sources of SARS-CoV-2 infection in humans. It is well-organized and thorough review- it covers potentially favorable conditions for the emergence of SARS-CoV-2, experimental models for SARS-CoV-2 infection, and importantly, animal species susceptible to SARS-CoV-2 infection. Overall, the review is well-written. Minor point- the authors are encouraged to check spelling, eg. "SAS-CoV-2" in the title.

Author Response

We thank the reviewer for the favorable comments.
We corrected the spelling of the name SARS-CoV-2 in the manuscript title.

Reviewer 2 Report

In this manuscript, Maurin and colleagues present a review about the possible animal sources of SARS-CoV-2 in humans. This a very interesting review and considering the dynamic, massive and accumulative research information about SARS-CoV-2 infection, an updated analysis described in this review is pertinent and necessary.

Specific comments:

  • Title correction: SARS-CoV-2 instead of SAS-CoV-2.
  • Line 113, explain briefly about furin cleavage site.
  • Authors do not mention other possible cell receptors besides ACE2.
  • Table 1. Reformulation of this table
  • Be specific and avoid redundant information. It may include the next titles per column: in silico, in vitro (in cells), in vivo. In each row, avoid the multiple use of terms like “susceptible to infection” or “not susceptible to infection”, “cell(s)”, instead you may include, for example: S=susceptible; NS=non-susceptible, below/under title “in vitro (in cell lines)”. Similar suggestion for in vivo, repetitive use of the term “susceptible”. Additionally, you may delete column 5 and add references at the end of each line, similarly, include citations for in silico and in vitro studies in each individual row.
  • Line 162, indicate reference.
  • Figure 1. No need to include this figure in the review, the legend figure information can be incorporated in the main text. Also, this figure may highlight the following connotation:
  • Racial, asian-like characters in the image, which is sensitive under the actual circumstances.
  • Line 228, indicate reference(s).
  • Line 415. Indicate the human ACE2 abbreviation: hACE2, in line 125 instead of line 415.
  • Delete “extra” periods in subtitles starting in 3.1.2.. Macaca mulatta (Rhesus macaque) until 3. . Captive wild animals in zoos. Also, do not forget italicized scientific names.
  • 8. Mice. This model has been used by various groups, please review and include more references.
  • 10. Pigs. Please, review the current status regarding the susceptibility of pigs to SARS-CoV-2 infection, for example, found in:

https://wwwnc.cdc.gov/eid/article/27/1/20-3399_article

https://www.ncbi.nlm.nih.gov/pmc/articles/PMC7594707/

Author Response

We are grateful to the reviewer for useful comments to improve our manuscript

Specific comments:

  • Title correction: SARS-CoV-2 instead of SAS-CoV-2.

We corrected the name spelling

  • Line 113, explain briefly about furin cleavage site.

The following sentences have been added (lines 122-125): “Although SARS-CoV-1 lacks the polybasic furin cleavage site found downstream of the RBD in the spike of SARS-CoV-2, such furin cleavage site (which confers a higher affinity of SARS-CoV-2 for the human ACE2 receptor), was described in many other lineages of coronaviruses and was naturally selected [30].”

  • Authors do not mention other possible cell receptors besides ACE2.

The following paragraph has been added to the section 2.2 (lines 179-183): “Beside ACE2, NRP-1 was reported to bind to furin-cleaved substrates, potentiating SARS-CoV-2 infectivity [76,77]. The ACE2 sequences from Mustelidae share about 83% amino acid identity with the human ACE2, while the ACE2 sequences from Neovison vison and Mustela lutreola have 99.51% similarity. In contrast, the NRP-1 protein is much more conserved among species (Devaux et al, manuscript under preparation).” The title of section 2.2 has also been changed to (line 141): “Interactions of viral spike with-ACE2 and other possible cell receptors.”

  • Table 1. Reformulation of this table

According to the reviewer’s suggestion, the following modifications have been made to Table 1:
- the title of the column “in vitro (in cells)” has been changed to “in vitro (in cells), data from [47,48], S=susceptible, NS=non susceptible, to SARS-CoV-2 infection”
- the title of the column “In vivo” has been changed to “in vivo, S=susceptible, NS=non susceptible, to SARS-CoV-2 infection”
These changes allow avoiding the multiple uses of the terms “susceptible to infection” or “not susceptible to infection”, as suggested by the reviewer.
- The references for “in silico” and “in vitro (in cells)” data have been cited the first line of the respective columns. It is not possible to spread these references by row because most studies have assessed the spectrum of hosts potentially susceptible to SARS-CoV-2 infection without focusing on a specific animal species.
- For “in vivo” data, the references have been added at the end of each line of column 4, and thus column 5 can now be deleted, as suggested by the reviewer.   

  • Line 162, indicate reference.

The appropriate references have been added (line 167).

  • Figure 1. No need to include this figure in the review, the legend figure information can be incorporated in the main text. Also, this figure may highlight the following connotation: Racial, asian-like characters in the image, which is sensitive under the actual circumstances.

We would prefer to keep Figure 1 in our manuscript if possible. A review with figures is more attractive to the reader. Nevertheless, on the advice of the reviewer, we have slightly modified this figure to reduce its possibly Asian character. The modified Figure 1 is presented in the revised version of the manuscript. Of course, it is not our intention to stigmatize anyone and we can further modify this figure if necessary. The new figure now replaces the old one in the text of the revised manuscript.

  • Line 228, indicate reference(s).

The appropriate references have been added (line 239).

  • Line 415. Indicate the human ACE2 abbreviation: hACE2, in line 125 instead of line 415.

This change has been done (line 136). “Human ACE2” has then been replaced by hACE2 in the rest of the text (lines 165, 190446, 447, and 451).

  • Delete “extra” periods in subtitles starting in 3.1.2.. Macaca mulatta (Rhesus macaque) until 3. . Captive wild animals in zoos. Also, do not forget italicized scientific names.

« Extra periods » have been deleted in the corresponding text section.

  • 8. Mice. This model has been used by various groups, please review and include more references.

The corresponding section has been extended to include more references. Some very recent studies have also been added. Lines 440 to 472.

  • 10. Pigs. Please, review the current status regarding the susceptibility of pigs to SARS-CoV-2 infection, for example, found in: https://wwwnc.cdc.gov/eid/article/27/1/20-3399_article
    https://www.ncbi.nlm.nih.gov/pmc/articles/PMC7594707/

Thank you for this suggestion. We have updated this paragraph using three new references, including the two you cite. Lines 501 to 523.

Please note that the sub-section “5.1 Transmission between animals” has been changed because two new cases of natural SARS-CoV-2 infection in wildlife minks have been recently reported in Spain. The following text has been added: “More recently, natural SARS-CoV-2 infection was further reported in two wild American minks in the Valencia Community, Spain [155]. SARS-CoV-2 Infection was confirmed by viral RNA detection in mesenteric lymph nodes. These animals were trapped far away from the nearest fur farm, suggesting other sources of infection, including SARS-CoV-2 contaminated wastewater.” The previous sentence has been deleted: “To our knowledge, this is the only SARS-CoV-2 infection in a wild animal.” Lines 731 to 736. 

We have updated the reference list to include all new references added to the revised manuscript. Reference [30] published in 2021 has also been updated.